# Genome-Wide Characterization of PEBP Gene Family and Functional Analysis of *TERMINAL FLOWER 1* Homologs in *Macadamia integrifolia*

**DOI:** 10.3390/plants12142692

**Published:** 2023-07-19

**Authors:** Jing Yang, Conghui Ning, Ziyan Liu, Cheng Zheng, Yawen Mao, Qing Wu, Dongfa Wang, Mingli Liu, Shaoli Zhou, Liling Yang, Liangliang He, Yu Liu, Chengzhong He, Jianghua Chen, Jin Liu

**Affiliations:** 1School of Life Sciences, Southwest Forestry University, Kunming 650224, China; yangjing3007@swfu.edu.cn (J.Y.); nch624@swfu.edu.cn (C.N.); liumingli08@163.com (M.L.); 2CAS Key Laboratory of Tropical Plant Resources and Sustainable Use, CAS Center for Excellence for Molecular Plant Science, Xishuangbanna Tropical Botanical Garden, Chinese Academy of Sciences, Kunming 650223, China; maoyawen16@mails.ucas.ac.cn (Y.M.); wuqing18@mails.ucas.ac.cn (Q.W.); wangdongfa@xtbg.ac.cn (D.W.); zhoushaoli@xtbg.ac.cn (S.Z.); yangliling@xtbg.ac.cn (L.Y.); heliangliang@xtbg.ac.cn (L.H.); oklahomaliu@163.com (Y.L.); 3Yunnan Institute of Tropical Crops, Jinghong 666100, China; liuziyan90@163.com (Z.L.); z_cheng.93@foxmail.com (C.Z.); 4University of Chinese Academy of Sciences, Beijing 100049, China; 5School of Life Sciences, University of Science and Technology of China, Hefei 230027, China

**Keywords:** PEBP family, *Macadamia integrifolia*, *MiTFL1*, juvenile period, plant architecture

## Abstract

Edible *Macadamia* is one of the most important commercial nut trees cultivated in many countries, but its large tree size and long juvenile period pose barriers to commercial cultivation. The short domestication period and well-annotated genome of *Macadamia integrifolia* create great opportunities to breed commercial varieties with superior traits. Recent studies have shown that members of the phosphatidylethanolamine binding protein (PEBP) family play pivotal roles in regulating plant architecture and flowering time in various plants. In this study, thirteen members of *MiPEBP* were identified in the genome of *M. integrifolia*, and they are highly similarity in both motif and gene structure. A phylogenetic analysis divided the *MiPEBP* genes into three subfamilies: *MFT-like*, *FT-like* and *TFL1-like*. We subsequently identified two *TERMINAL FLOWER 1* homologues from the *TFL1-like* subfamily, *MiTFL1* and *MiTFL1-like*, both of which were highly expressed in stems and vegetative shoots, while *MiTFL1-like* was highly expressed in young leaves and early flowers. A subcellular location analysis revealed that both *MiTFL1* and *MiTFL1-like* are localized in the cytoplasm and nucleus. The ectopic expression of *MiTFL1* can rescue the early-flowering and terminal-flower phenotypes in the *tfl1–14* mutant of *Arabidopsis thaliana*, and it indicates the conserved functions in controlling the inflorescence architecture and flowering time. This study will provide insight into the isolation of *PEBP* family members and the key targets for breeding *M. integrifolia* with improved traits in plant architecture and flowering time.

## 1. Introduction

The phosphatidylethanolamine binding protein (PEBP) family is ancient, and the encoded protein sequences are highly conserved, being present in plants, animals and microorganisms [1,2,3]. In plants, the *PEBP* family can be divided into three subfamilies: *MOTHER OF FT AND TFL1-like* (*MFT-like*), *FLOWERING LOCUS T-like* (*FT-like*) and *TERMINAL FLOWERING 1-like* (*TFL1-like*), which are involved in both the regulation of flowering time and the control of plant architecture [4,5]. *MFT-like* genes are the ancestors of *FT-like* and *TFL-like* genes, and both *FT-like* and *TFL1-like* genes are found only in flowering plants now, while *MFT-like* genes are also present in lower mosses and lycophytes [3,6,7].

The functions of these *PEBP* genes have been comprehensively studied in Arabidopsis thaliana. There are six *PEBP* genes identified in Arabidopsis: *MOTHER OF FT AND TFL1* (*MFT*), *FLOWERING LOCUS T* (*FT*), *TWIN SISTER OF FT* (*TSF*), *BROTHER OF FT AND TFL1* (*BFT*), *TERMINAL FLOWER* 1 (*TFL1*) and *ARABIDOPSIS THALIANA CENTRORADIALIS* (*ATC*) [8]. *MFT* belongs to the *MFT-like* subfamily and is mainly expressed in seeds [9]. As a weak floral inducer, the overexpression of *MFT* promotes early flowering in Arabidopsis [10]. In addition to its role in regulating flowering time, *MFT* also regulates seed germination via the ABA and GA signaling pathways [9,11] and through BR, partly against ABA, to regulate seed germination and fertility [12]. Both *FT* and *TSF* belong to the *FT-like* subfamily, and their functions largely overlap in promoting flowering time [5]. *TSF* is mainly expressed in the hypocotyl and petiole vasculature, while *FT* is mainly expressed in the cotyledons and leaves [13]. *FT* acts as a floral activator which is transported from the leaf to the shoot apical meristem and then interacts with another floral regulator, *FLOWERING LOCUS D* (FD), to promote flowering via the photoperiod and temperature pathways [14,15]. The overexpression of *FT* results in early flowering [8]. In addition to promoting flowering, *FT* and *TSF* also affect the stomatal opening [16]. The *TFL1-like* subfamily, including *BFT*, *TFL1* and ATC, all repress flowering [17]. *BFT* is expressed in the shoot apical meristem, young leaf, and axillary inflorescence meristems [18]. The overexpression of *BFT* leads to the development of abnormal inflorescence, as well delays in flowering time [18]. The expression of ATC was detected only in the hypocotyls of young plants but not in the inflorescence meristem [17]. ATC is a short day (SD)-induced floral inhibitor that moves long distances, interacts with FD and antagonizes *FT* to affect flowering [14]. *TFL1* is expressed in the SAM and axillary bud meristems to maintain vegetative phase growth, influencing flowering time and plant architecture [19]. The tfl1 mutants flower earlier and produce fewer leaves, shoots and flowers, and the SAM then converts into a terminal flower at the late developmental stages in long days [19,20]. In Arabidopsis, the overexpression of *TFL1* produces an expanded vegetative rosette and a highly branched inflorescence and greatly extends the vegetative period [21]. In previous studies, *TFL1* competes with *FT* for binding to FD to repress the key flowering genes LFY and AP1, thereby delaying the flowering time [22].

Unlike annual herbaceous *Arabidopsis*, perennial trees experience much longer juvenile periods, often lasting years to decades before first acquiring flowering capacity; during the subsequent adult period, they show seasonal alternations between vegetative and reproductive growth [23]. Moreover, trees exhibit huge and complex architectures. *TFL1* is a major floral repressor that maintains meristem indeterminacy and also has a strong impact on plant architecture [24,25,26]. In plants with longer juvenile stages, the reduced expression of *TFL1* accelerates flowering [27]. The woody biofuel plant *Jatropha curcas* can flower around the age of nine months, while transgenic plants overexpressing three *JcTFL1* genes showed an extreme late flowering phenotype that did not flower after having been planted for three years, and *JcTFL1b* RNAi plants flowered three months earlier than wild-type plants [28]. The *JcTFL1* genes also affected morphology and architecture in *J. curcas*. Lines overexpressing *JcTFL1s* did not exhibit branching in the first year after planting. European pear has a long juvenile period, and the silencing of *PcTFL1-1* and *PcTFL1-2* via RNAi greatly shortened the juvenile period and early flowering in lines [29]. The RNAi transgenic line of pear is small in size, and the apical meristems eventually terminate with flowers, while one or more lateral buds continue to grow. Similarly, silencing the *MdTFL1* gene in apple reduces the juvenile phase and generation time, causing the trees to be less vigorous, with shorter branches [30,31]. Mutations in the *KSN* (*TFL1* homologues) gene affected the flowering time of rose and greatly shortened the vegetative phase [32]. *PopCEN1* belongs to the *TFL1-like* subfamily that maintains poplar shoot meristem identity, and the flowering time occurs early in *PopCEN1*-RNAi trees [33]. The downregulation of *PopCEN1* and its paralog *PopCEN2* also affect the inflorescence number and the short branch proportion. Thus, *TFL1* controls several aspects of plant development, including the juvenile phase, flowering time, inflorescence architecture, shoot growth pattern and life history strategies [27]. Given that woody crops generally have large architectures and long juvenile periods, they are not conducive to commercial production. *PEBP* family genes have potential value for breeding optimal flowering times and ideal plant types.

The genus *Macadamia F. Muell.* belongs to the *Proteaceae* family [34]. Among four common nut species of this genus, *M. ternifolia* and *M. jansenii* are small trees that produce inedible nuts, whereas *M. integrifolia, M. tetraphylla* and their hybrids are cultivated worldwide for their edible premium kernels [35]. *Macadamia* nuts have a unique flavor and are rich in lipids, proteins and important micronutrients and are favored by consumers [36,37]. *M. integrifolia* is a dicotyledon plant with a genome (2n = 28) size of about 652–896 Mb [38]. At present, the genome sizes of two cultivars are known: 745 Mb for HEAS 741 [39] and 794 Mb for HEAS 344 (Kau) [40]. The domestication history of *Macadamia* is short, the record is clear, and the release of genome data provides the possibility of improving the efficiency of *Macadamia* breeding [40]. In this study, we first identified all potential *PEBP* family genes of *M. integrifolia* and systematically analyzed their phylogenetic relationships, chromosome locations, gene structures, motifs and promoter cis-acting elements. In other species, *TFL1* was reported to influence the first flowering time and plant architecture. To understand the function of the *MiTFL1* gene in *M. integrifolia*, we analyzed the expression pattern of *MiTFL1* and its ectopic expression in *Arabidopsis* to verify its function. This study will provide a reference for understanding the different roles of *PEBP* family members in flowering time and plant architecture regulation in *M. integrifolia*.

## 2. Results

### 2.1. Identification of the PEBP Family in M. integrifolia

A total of 13 *PEBP* genes were identified in the whole genome of *M. integrifolia* by combining an HMM and a BLASTP search. These 13 *PEBP* genes encode 16 transcripts (Table 1) because *MiMFT3*, *MiFT1* and *MiFT3* have two transcripts each due to variable splicing. The *M. integrifolia PEBP* family proteins range from 172 to 207 amino acids in length and from 18.98 to 23.12 kDa in molecular weight (MW), with a minimum protein isoelectric point (PI) of 5.87 and a maximum of 9.47 (Table 1). The Grand average of hydropathicity (GRAVY) values of less than 0 indicate that all *MiPEBPs* are hydrophilic (Table 1).

### 2.2. Gene Structure, Conserved Motif and Chromosomal Location Analysis of MiPEBP Genes

Most of the 13 *PEBP* family genes in *M. integrifolia* contain four exons and three introns except for *MiMFT3* and *MiFT3*, both of which contain five exons and four introns. Among them, the *MiFT3* gene is longer than the other genes, with a length of more than 65,000 bp (Figure 1A). A total of six conserved motifs were identified in the family of *MiPEBP* proteins. The motifs 1 to 5 were observed in almost all *MiPEBP*, and the sequences were consistent, indicating that *MiPEBP* genes are relatively conserved. Motif 6 was only present in some genes of the *MFT-like* and *TFL1-like* subfamilies but was not present in the *FT-like* subfamily (Figure 1B). In Figure 1C, the *MiPEBP* genes were mainly distributed on seven chromosomes, which were chromosomes 01, 03, 05, 06, 08, 10 and 11, respectively, and the distribution was uneven. Chromosomes 03, 06 and 08 contained two *MiPEBP* genes, while the remaining chromosomes contained only one gene for each. In addition, *MiFTL1-like* and *MiFT1* genes were located on two unanchored scaffolds.

### 2.3. Phylogenetic Analysis and Classification of the PEBP Gene Family in M. integrifolia

To investigate the relationships among the *MiPEBP* proteins, a phylogenetic tree was constructed via the neighbor joining method based on a multiple-sequence alignment of 126 sequences from *Oryza sativa*, *Zea mays*, *Brachypodium distachyon*, *Sorghum bicolor*, *Arabidopsis thaliana*, *Vitis vinifera*, *Solanum lycopersicum*, *Malus domestica* and *Macadamia integrifolia*. These 126 PEBP protein sequences were found to be classified into *MFT-like*, *FT-like* and TFL-like subfamilies (Figure 2A). Each species contains a different number of *PEBP* genes (Figure 2A). Compared to dicotyledonous plants, monocotyledonous plants exhibit higher numbers of *PEBE* genes (Figure 2B). In *M. integrifolia*, the *MFT-like* subfamily harbors three genes, including *MiMFT1*, *MiMFT2* and *MiMFT3*. Five genes were clustered into the *FT-like* subfamily, which were *MiFT1*, *MiFT2*, *MiFT3*, *MiFT4* and *MiFT5*. There were five genes including *MiTFL1-like*, *MiTFL1*, *MiBFT1*, *MiBFT2*s and *MiBFT3* identified in the TFL-like subfamily (Figure 2A).

### 2.4. Analysis of Cis-Acting Regulatory Elements in Promoter Regions of MiPEBP Genes

An analysis of 3000 bp promoter sequences on the upstream of each of the 13 *MiPEBP* genes revealed that there are multiple types of cis-acting elements which are involved in light-responsive, stress-responsive and hormone-responsive related elements. Among these regulatory elements, light-responsive elements are the most abundant (Figure 3). The stress-responsive elements include drought stress response, low temperature response, wounding response, anaerobic induction, defense and stress response elements (Figure 3). Hormone-responsive elements are also abundant, and the number of responsive elements ranged from the highest to lowest ranked as the MeJA-responsive element, abscisic acid responsive element, gibberellin responsive element, auxin responsive element and salicylic acid responsive element (Figure 3). Results of the promoter cis-acting element analysis indicated that the *PEBP* gene family in *M. integrifolia* plays important roles in hormonal regulation, responses to light signals and resisting abiotic stress.

### 2.5. Identification and Multiple Alignment of MiTFL1 Homologue

To further analyze members of the *TFL1-like* subfamily of *M. integrifolia*, proteins which are homologous to the *Arabidopsis TFL1-like* subfamily and the *Macadamia TFL1-like* subfamily were selected to construct the phylogenetic tree. The results showed that the *Macadamia TFL1-like* members are mainly classified into two clades. Two are homologous to the *TFL1,* while three are homologous to the *BFT* (Figure 4A).

A multiple sequence alignment of the *MiTFL1*s and other *TFL1* homologs showed that the key amino acid residues His88 and Asp144 in *TFL1*, which lead to the functional divergence between *TFL1* and *FT* in *Arabidopsis*, are quite conserved in *MiTFL1* and *MiTFL1-like* (Figure 4B). Two motifs are highly conserved among proteins of the *PEBP* family, D-P-D-X-P and G-X-H-R, which contribute to the conformation of the *PEBP* family. Ligand binding site motifs are also present in exons 2 and 4 of *MiTFL1* and *MiTFL1-like* (Figure 4B). Exon 4 plays a key role in the function of *FT*/*TFL1* proteins, and the B and C segments are particularly important in the determination of the functional specificity of *FT* and *TFL1*. Unlike *FT*, the amino acids in segment B evolved rapidly between *TFL1* homologs and show similarities between *MiTFL1* and *MiTFL1-like* (Figure 4B).

### 2.6. Subcellular Localizations of MiTFL1 and MiTFL1-like

To detect the subcellular localizations of *MiTFL1* and *MiTFL1-like*, *35S::MiTFL1-GFP* and *35S::MiTFL1-like-GFP* fusion plasmids were constructed and transiently expressed in tobacco leaves. The GFP signals of *MiTFL1*and *MiTFL1-like* were distributed in both the nucleus and cytoplasm and were detected via confocal microscopy. The green fluorescent protein of the control group was distributed in the nucleus and the cytoplasm (Figure 5). These results suggest that both *MiTFL1* and *MiTFL1-like* proteins are localized in the cytoplasm and nucleus to function.

### 2.7. Expression Patterns of MiTFL1 and MiTFL1-like in M. integrifolia

To check the expression patterns of *MiTFL1* and *MiTFL1-like* genes, different tissues of *M. integrifolia* trees which were about 10 years old and grown in Jinghong, Yunnan Province, were collected (Figure 6A–I), and the transcripts of the two genes were detected via real-time quantitative PCR. Compared with the floret- and fruit-related tissues, *MiTFL1-like* is mainly expressed in the vegetative shoot, stem and young leaf (Figure 6J). It is noteworthy that the transcript of *MiTFL1* is highly and specifically detected in the vegetative shoot and stem and is hardly detected in the racemes and florets of different stages (Figure 6K). This indicates that the functional diversity between *MiTFL1* and *MiTFL1-like* may occur at the transition from the vegetative to the reproductive development stage in *M. integrifolia*.

### 2.8. The Role of MiTFL1 in Regulating Flowering Time

The *35S::MiTFL1* transgenic lines of *Arabidopsis,* which were obtained via the flower dip transformation of a *tfl1–14* mutant, were planted to analyze the plant phenotypes. In contrast to the *Arabidopsis tfl1–14* mutants, several *MiTFL1* transgenic lines did not produce visible flower buds until 15 days after the flowering of *tfl1–14* mutants, and several transgenic lines of *MiTFL1* never produced visible flower buds (Figure 7A). These transgenic lines flowered significantly later compared to the *tfl1–14* mutant (Figure 7B). The flowering time line 187# was significantly later than the flower time of the wild-type (Figure 7B). The transgenic lines showed more rosette leaves than the mutant *tfl1–14* (Figure 7C), indicating that the transgenic lines underwent longer vegetative periods.

### 2.9. The Phenotypic Analysis of MiTFL1 in Arabidopsis

Compared with the *tfl1–14* mutant of *Arabidopsis*, *35S::MiTFL1* transgenic lines had higher main stem heights, longer lateral branches and more lateral branches on the main stem, indicating that the transformation of *MiTFL1* essentially restored the mutant phenotypes of *tfl1–14* (Figure 8). The meristems of the transgenic lines exhibited a longer vegetative period similar to that of the wild-type, which was different from the terminal flower phenotype of the *tfl1–14* mutant (Figure 8). These results indicate the essential function of *MiTFL1* in maintaining meristem activity during the vegetative phase, which is similar to that of *TFL1* in *Arabidopsis*. Meanwhile, some transgenic lines displayed abnormal flowers and siliques, with sepal and petal fusion, pedicel elongation and lack of silique fullness (Figure 8).

## 3. Discussion

Flowering is a key process which indicates the transition from vegetative to reproductive growth for plants, and genes of the *PEBP* family play important roles during this process and in the final plant architecture [3]. Up to now, the genes of the *PEBP* family have been identified in many plant species, such as quinoa, perilla, sugarcane, rice, garlic, etc., and the number of *PEBP* genes varies among these species [41,42,43,44,45]. A total of 13 *PEBP* family members were identified in *M. integrifolia*, and they were classified into three subfamilies: *MFT-like*, *FT-like* and *TFL1-like*. Further duplications of the *PEBP* genes may occur as flowering plants evolve, which would give rise to a varied number of *PEBP* genes among species. Dicot species tend to have fewer *PEBP* genes than monocot species. In the phylogenetic tree, the *FT-like* subfamily of dicots showed less branches than that of monocots, suggesting that the function of the *FT* gene may be more complex in monocots. In most *MiPEBP* family members, the gene structure is very conserved, with four exons and three introns. Almost all *MiPEBP* members contain motifs 1 to 5, which are arranged in a consistent order. This illustrates that members of *MiPEBP* family are relatively conserved in sequence similarity, consistent with the high conservation of the *PEBP* gene among different species. The complex physiological process of flowering is regulated by photophore, hormone biosynthesis and signaling [46]. An analysis of the promoter elements of the *MiPEBP* family revealed the presence of a large number of different types of cis-acting elements on their promoter regions, with an abundance and high number of light- and hormone-responsive element types, implying that they may play important roles in regulating the photoperiod pathway and gibberellin pathway of plant flowering. Thus, the characterization of *PEBP* gene functions is a key means of fine-tuning the flowering of *M. integrifolia*.

Functional studies of *PEBP* genes have been widely reported in different species. *FT* and *TFL1* are two important genes acting downstream of the flowering regulation network [47]. *FT* and *TFL1* have highly similar amino acid sequences but have antagonistic functions for flowering time in plants [48]. This functional difference is only caused by the differentiation of one critical amino acid residue and a conserved amino acid segment in the PEBP domain [48]. Several other genes have been reported to regulate flowering. Within the SAM domain, *TFL1* forms a complex with the bZIP transcription factor FD and 14-3-3 proteins to repress the expression of the flowering-time-related genes *LFY*, *AP1* and *CAL*, thereby repressing the floral transition [22]. The photoperiod signal is transmitted to *FT* by *CO*, and then the *FT* protein begins to translocate from the leaf to the SAM [49]. Similar to *TFL1*, *FT* competitively binds FD and activates the downstream flowering-related genes *SOC1*, *AP1* and *LFY* to promote flowering [50]. In addition to delaying flowering time, *TFL1* influences the growth habit and inflorescence architecture in plants. The *TFL1-like* gene *CENTRORADIALIS* (*CEN*) of *Antirrhinum majus* was the first *PEBP* gene identified in plants. The flowering time in the *cen* mutant was unaffected, but the plants were short and compact, and the inflorescences terminated early [51]. Rice (*Oryza sativa*) has four *TFL1* homologous: *RCN1*, *RCN2*, *RCN3* and *RCN4*. *RCN1* and *RCN2* delay flowering and increase the number of tillers [52]. Similarly, knocking out *RCN* genes in rice will lead to smaller panicles and fewer branches [53]. In soybean (*Glycine max*), *GmTFL1b/Dt1* is involved in the stem determinate growth [54]. The tomato *TFL1* ortholog *SP* regulates the indeterminate growth habit of the apical meristem and delays flowering time [55]. All these comparative studies indicate the conserved function of *TFL1* in the inflorescence architecture and flowering among different plant species. The expression of *MiTFL1* genes in different tissues of *M. integrifolia* was analyzed. The expression patterns of *MiTFL1* and *MiTFL1-like* were similar, and both were expressed in the stem and vegetative shoot. *MiTFL1-like* is also expressed in young leaves and early flowers, suggesting that *MiTFL1-like* may play other functions during flower and leaf development. All transgenic *Arabidopsis* lines which overexpressed *MiTFL1* showed delayed anthesis and indeterminate inflorescence. Based on the above studies, we have confirmed that there are significant differences in expression patterns between *MiTFL1* and *MiTFL1-like*, with *MiTFL1* having a conserved function in flowering regulation and plant architecture. In view of the conservative function of the *TFL1* gene in flowering regulation and plant architecture, other *Macadamia* cultivars may also show similar expression patterns and functions.

*PEBP*-family-related genes mainly play important roles in regulating seed dormancy, flowering time and plant architecture [3,56], and can solve many problems in the actual production process, such as improving seed germination rate, prolonging the vegetative growth period of vegetable crops, prolonging the flowering period of ornamental plants and shortening the juvenile state of woody plants in breeding. In agricultural production, the *sp* mutant of tomato has been widely used in tomato breeding because of its shorter vegetative growth period, earlier flowering period and earlier fruit maturity [24]. Cucumber plants with function deletions of the *CsTFL1* gene showed significantly limited vegetative growth, and the plant height was greatly reduced, which was more convenient for management and had great application potential [57]. Different from herbaceous crops, *Macadamia* is a large subtropical rainforest tree which originates from Australia [58]. Cultivated *Macadamia* nuts grow at an average height of 12–15 m and require an average of 15 years to reach peak yield [58,59]. Both the large plant size and long juvenile period hamper commercial production and breeding efficiency [35,60]. Large trees require pruning to maintain productivity, while the small tree variety enables high planting density and early production [35]. Traditional methods to breed new *Macadamia* cultivars are time-consuming, laborious and expensive [61]. Genetic engineering of *Macadamia* will accelerate the development of new cultivars with excellent agronomic traits [62]. This research indicates that the *PEBP* genes have potential advantages for breeding valuable *Macadamia* cultivars with early flowering times and commercial plant architectures.

## 4. Materials and Methods

### 4.1. Plant Materials and Growth Conditions

The *M. integrifolia* (HEAS 863) trees used in this study were planted in a *Macadamia* germplasm base of the Yunnan Institute of Tropical Crops. Different organs from the stem, leaf, vegetative shoot, inflorescence, flower and nutlet were collected, placed in liquid nitrogen, and stored at −80 °C for RNA extraction.

*Arabidopsis thaliana* wild type Col-0 and *tfl1–14* mutant seeds were sown after 3 days of treatment at 4 °C. *Nicotiana benthamiana* seeds were sown on wet soil. *Arabidopsis* and tobacco grew in a greenhouse under a 16/8 h light (150 μE m^−2^ s^−1^)/dark cycle and 24/20 °C conditions at a relative humidity of 50–60%; the plants were well-watered and received adequate nutrition.

### 4.2. Identification of PEBP Family Genes in M. integrifolia

In order to identify *PEBP* genes from *M. integrifolia*, the annotated genome data of *M. integrifolia* were downloaded from GenBank (https://www.ncbi.nlm.nih.gov/genbank/, GCF_013358625.1, accessed on 15 June 2022). The hidden Markov model (HMM) profile of the PBP domain (PF01161) was obtained from the Pfam database and used as the query; then, *M. integrifolia* amino acid sequences were also searched using HMMER v3.3.2, E-value ≤ 10^−5^. The protein sequences of six *PEBP* members of *A. thaliana* were downloaded as query sequences from the *Arabidopsis* database (https://www.arabidopsis.org/, accessed on 15 June 2022). BLASTP was performed with the protein sequences of *M. integrifolia*, E-value ≤ 10^−5^. Candidate sequences obtained via the two methods were uploaded to the Pfam website (http://pfam-legacy.xfam.org/, accessed on 17 June 2022) for further comparison and screening based on the PEBP domain, and it was confirmed that all members contained conserved PEBP domains. The physical and chemical parameters of the PEBP proteins, including the theoretical isoelectric point (PI), molecular weight (MW) and grand average of hydropathicity (GRAVY), were determined using ProtParam (https://web.expasy.org/protparam/, accessed on 6 August 2022).

### 4.3. Gene Structures, Protein Motifs and Chromosome Locations

The exon and intron locations of the *PEBP* genes were analyzed by comparing the coding sequences with their genome sequences. The Multiple Em for Motif Elicitation (MEME) online tool (https://meme-suite.org/meme/tools/meme, accessed on 15 August 2022) was used to predict protein motifs. The chromosome distributions of *PEBP* genes were obtained based on genome GFF3 files. Finally, the gene structures, protein motifs and chromosome locations were visualized with the software TBtools (v1.120) [63].

### 4.4. Phylogenetic Analyses and Multiple Alignments

Phylogenetic trees were generated by using MEGA 7.0 with the neighbor-joining (NJ) algorithm. Bootstrap values from 1000 replications were used to assess group support, and the substitution model used was the Poisson model. The phylogenetic tree was visualized with the iTOL tool (https://itol.embl.de, accessed on 2 May 2023). Multiple alignments of the sequences were performed using the DNAMAN software. Gene accession numbers are listed in Appendix A.

### 4.5. Analysis of Cis-Acting Regulatory Elements in Promoter of MiPEBPs

The upstream 3000 bp promoter sequences of 13 *MiPEBP* genes were uploaded to the PlantCARE Database (http://bioinformatics.psb.ugent.be/webtools/plantcare/html/, accessed on 12 March 2023) to search for cis-acting elements. The physical distribution of the various cis-acting elements was visualized in TBtools (v1.120) [63].

### 4.6. RNA Extraction and qRT-PCR

Total RNA from *M. integrifolia* plants was extracted via a modified CTAB method, and total RNA from *Arabidopsis* plants was extracted using a RnaEx™ Total RNA isolation kit (GENEray Biotech, Shanghai, China). The RNA was quantified using a NanoDrop 2000 spectrophotometer (Thermo Fisher Scientific, Shanghai, China), and the quality was checked via electrophoresis. The first-strand cDNA was reverse-transcribed with the HiScript^®^II 1st Strand cDNA Synthesis Kit (+gDNA wiper) (Vazyme, Nanjing, China). A quantitative real-time PCR was performed using Magic SYBR Green qPCR Mix (Magic-bio, Hangzhou, China) on a LightCylcer 480 device (Roche, Basel, Switzerland). *AtActin* and *MiActin* were used as internal reference genes. The data were calculated using the 2^−ΔΔCt^ method. Primer sequences are listed in Appendix A.

### 4.7. Vector Construction and Plant Transformation

The full-length coding sequence of *MiTFL1* was amplified via PCR from the cDNA of vegetative shoots with Phanta MaxSuper-Fidelity DNA Polymerase (Vazyme, Nanjing, China). All PCR products were detected via 1.5% agarose gel electrophoresis and purified using an EasyPure^®^ Quick Gel Extraction Kit (TransGen, Beijing, China). The PCR-amplified fragment products were inserted into the pCAMBIA3301 vectors between the *Nco*I and *Bst*EII sites via a ClonExpress II One Step Cloning Kit (Vazyme, Nanjing, China) under the CaMV35S promoter. The recombinant plasmid was transformed into the Agrobacterium tumefaciens EHA105 strain for transformation into the *A.* thaliana *tfl1–14* mutant. *Arabidopsis* plants were transformed via the floral dip method [64]. T0 transgenic seeds were sown in the soil and then selected by spraying Basta. The identification of T1 transgenic plants is shown in Appendix A.

### 4.8. Subcellular Localization

The CDS sequences of *MiTFL1* and *MiTFL1-like* were amplified and connected to the pPYS22-GFP vector between the *Xho*I and *Kpn*I sites via homologous recombination. The recombinant plasmids were transferred into *Agrobacterium tumefaciens* strain EHA105. They were centrifuged after overnight incubation and resuspended as an *Agrobacterium* pellet in an infiltration solution (10 mM of MgCl_2_, 10 mM of MES and 100 µM of acetosyringone, PH = 5.6) to a desired optical density (OD600 =1.0). The 4-week-old tobacco leaves were infected with the mixed solution. After 2 days in the dark, the GFP signal was observed via a laser confocal microscope and the empty vector was used as a control.

## Figures and Tables

**Figure 1 plants-12-02692-f001:**
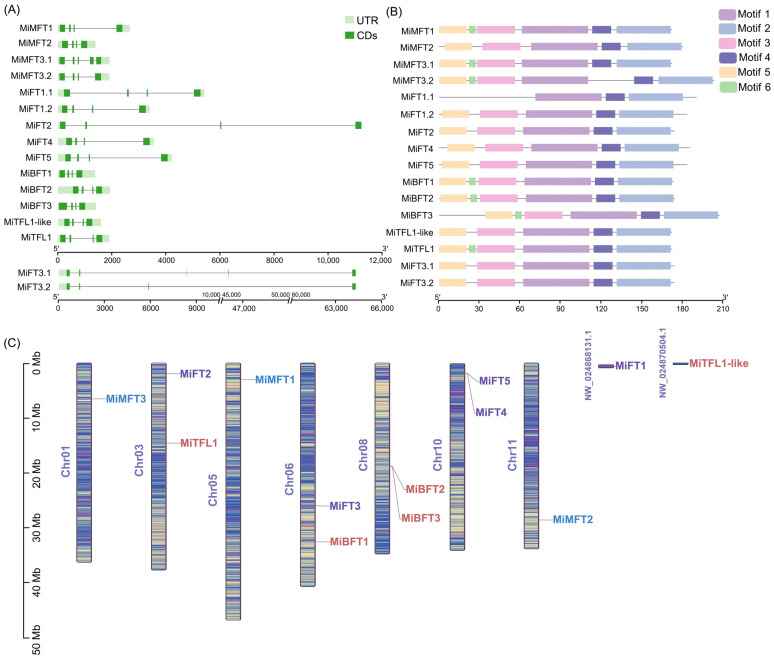
Gene structures, conserved motifs and chromosomal locations of *PEBPs* in *M. integrifolia*. (**A**) The gene structures of *MiPEBP* genes. (**B**) The conserved motifs distributed in *MiPEBP* genes. (**C**) The chromosomal locations of the *MiPEBP* genes.

**Figure 2 plants-12-02692-f002:**
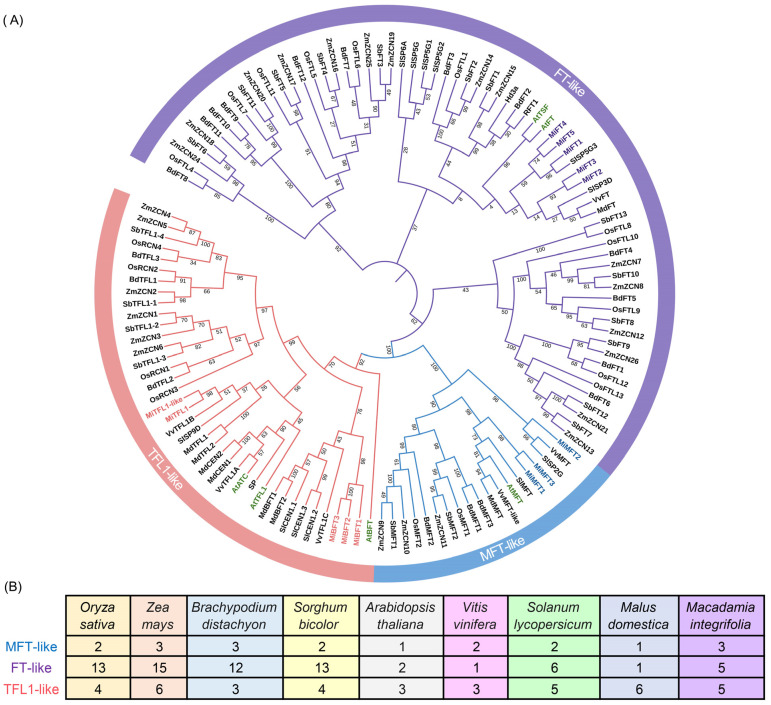
The phylogenetic relationship and the gene numbers of the *PEBP* genes. (**A**) Phylogenetic relationship and (**B**) the gene numbers of the *PEBP* family in *O. sativa* (*Os*), *Z. mays* (*Zm*), *B. distachyon* (*Bd*), *S. bicolor* (*Sb*), *A. thaliana* (*At*), *V. vinifera* (*Vv*), *S. lycopersicum (Sl), M. domestica* (*Md*) and *M. integrifolia* (*Mi*).

**Figure 3 plants-12-02692-f003:**
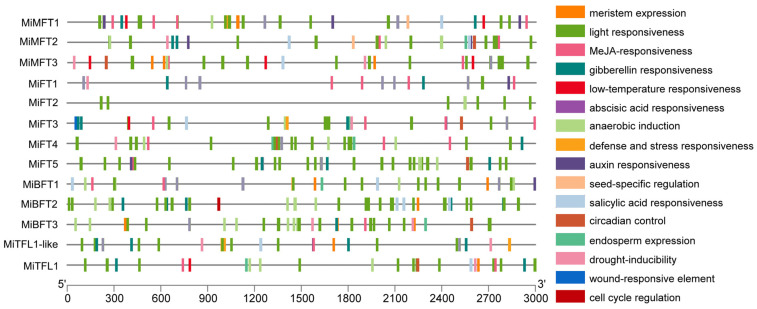
Cis-acting elements among the promoter regions of *MiPEBP* family members. The different colored boxes represent various cis-acting elements located in the promoter regions of *MiPEBP* members.

**Figure 4 plants-12-02692-f004:**
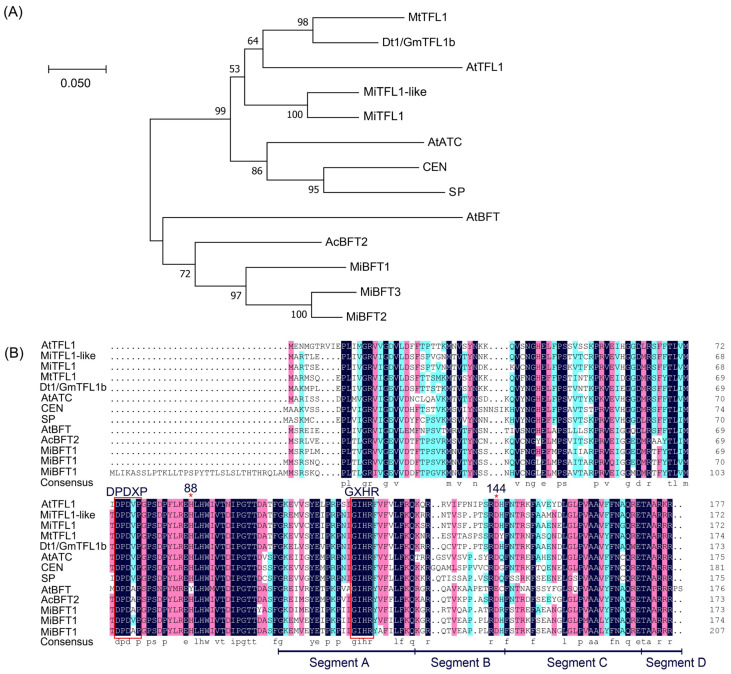
Identification and multiple alignment analysis of *TFL1* proteins. (**A**) Phylogenetic relationship of the *TFL1* subfamily. (**B**) Multiple sequence alignment analysis between *MiTFL1* proteins and other *TFL1* homologues.

**Figure 5 plants-12-02692-f005:**
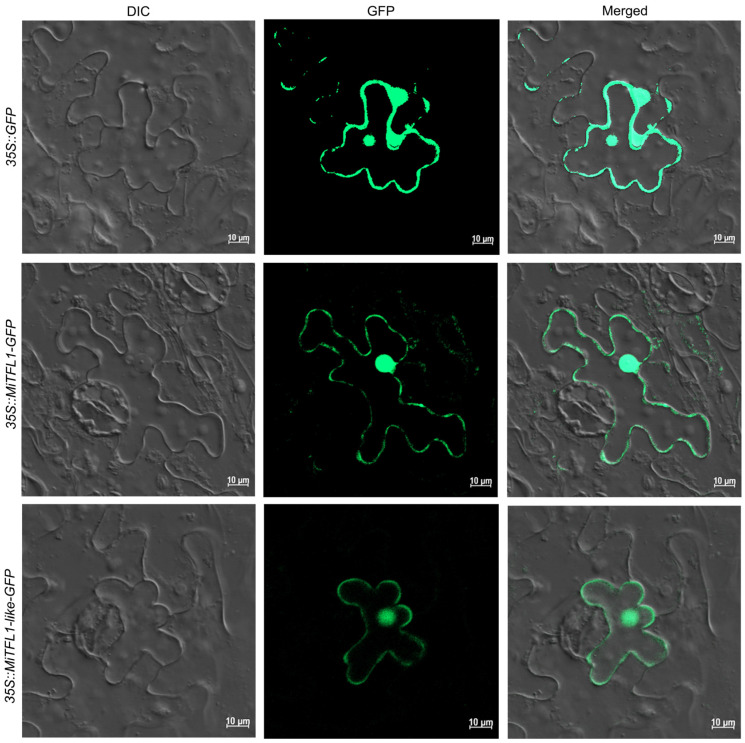
The subcellular localizations of *MiTFL1* and *MiTFL1-like* in the epidermal cells of tobacco leaves. The GFP signals of the *MiTFL1*-GFP and *MiTFL1-like*-GFP fusion proteins appeared in the nucleus and cytoplasm. The free GFP was driven by CaMV35S promoter as a control.

**Figure 6 plants-12-02692-f006:**
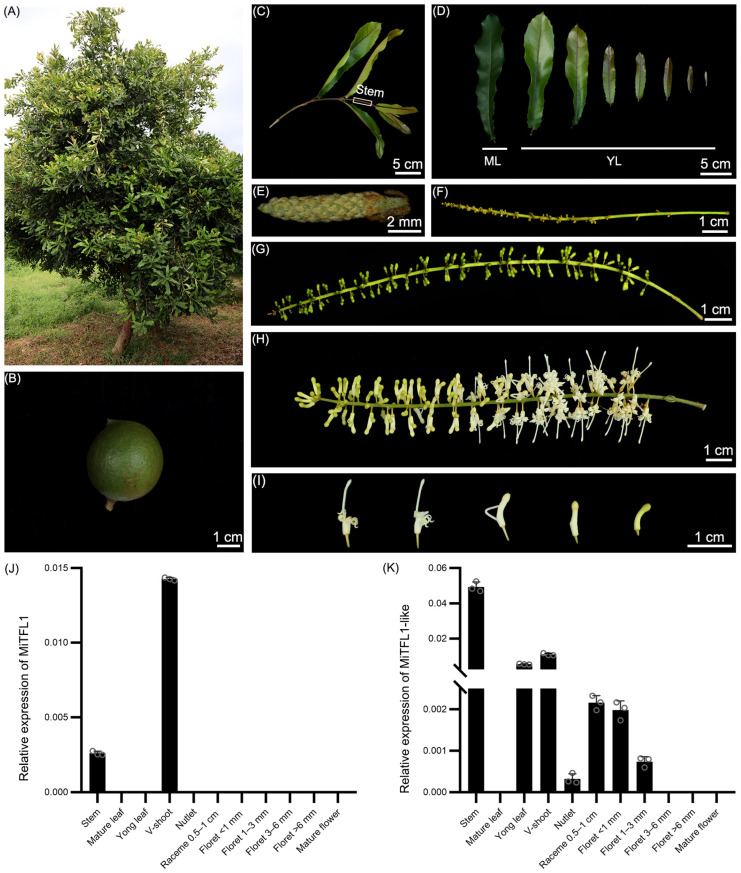
Morphologies of different organs and expression patterns of *MiTFL1* and *MiTFL1-like* in different tissues of *M. integrifolia.* (**A**) A ten-year-old *M. integrifolia* tree. (**B**) A mature *M. integrifolia* nut. (**C**) A branch of *M. integrifolia* tree containing a vegetative shoot (V-shoot) and a young stem. (**D**) The leaves at different development stages of *M. integrifolia*. ML—mature leaf; YL—young leaf. (**E**) Raceme, about 1 cm. (**F**–**H**) Racemes of different lengths with florets in different periods. (**I**) Florets in different periods, collected from different part of racemes. (**J**) Relative expression of *MiTFL1*. (**K**) Relative expression of *MiTFL1-like*.

**Figure 7 plants-12-02692-f007:**
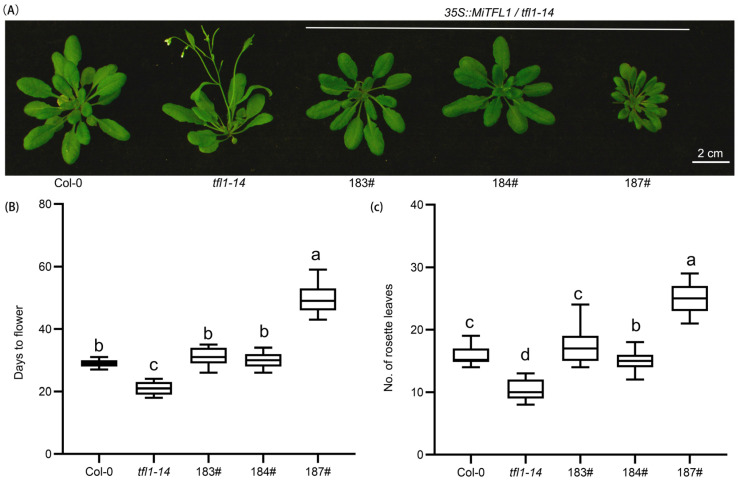
The flowering time analysis of transgenic *Arabidopsis*. (**A**) The phenotypes of *MiTFL1* transgenic lines 10 days post anthesis for *tfl1–14.* (**B**) The days to flower and (**C**) the number of rosette leaves. Data were collected at bolting time with 15 plants per line (Appendix A). In (**B**,**C**), a, b, c, d, different letters indicate statistically significant differences between each other. The comparison was conducted via a one-way analysis of variance (ANOVA), and a significance level of *p* < 0.05 was considered as statistically significant.

**Figure 8 plants-12-02692-f008:**
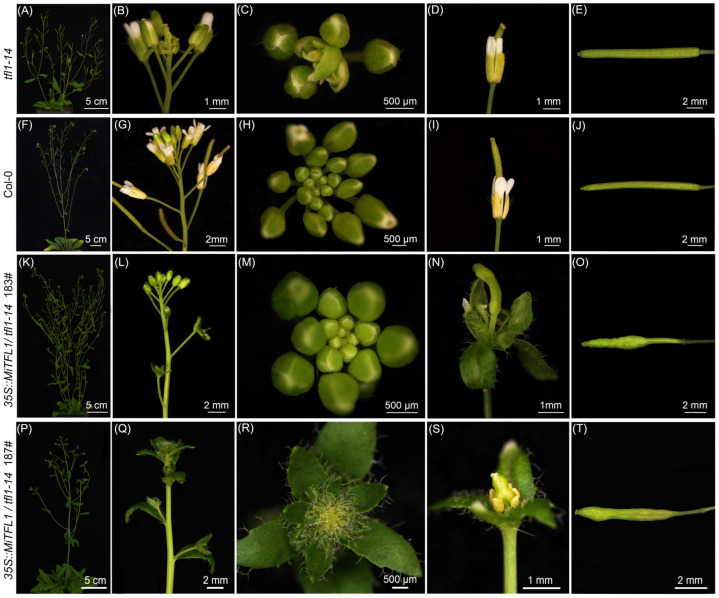
Phenotypic traits of ectopically overexpressed *MiTFL1* in transgenic *Arabidopsis*. Flowering plants of *tfl1–14* (**A**), Col-0 (**F**) and the transgenic lines 183# (**K**) and 187# (**P**). Close-up images of inflorescence in *tfl1–14* (**B**), Col-0 (**G**) and the transgenic lines 183# (**L**) and 187# (**Q**). The close-up images of inflorescence apex in *tfl1–14* (**C**), Col-0 (**H**) and the transgenic lines 183# (**M**) and 187# (**R**). Normal flowers in *tfl1–14* (**D**) and Col-0 (**I**). Abnormal flowers in 183# (**N**) and 187# (**S**) *transgenic Arabidopsis*. Normal siliques in *tfl1–14* (**E**) and Col-0 (**J**) and abnormal siliques in 183# (**N**) and 187# (**S**) *transgenic Arabidopsis*. The terminal-flower in *tfl1–14* (**B**,**C**). The vegetative shoot maintains characteristics similar to the vegetative bud for a long time (**R**), and a new inflorescence structure will grow at the position where the flower should grow (**Q**). The flower morphology is abnormal: sepals and petals fuse, and sometimes the fruit stalk elongates during the development of siliques (**N**,**S**). The unfilled siliques (**O**,**T**).

**Table 1 plants-12-02692-t001:** Detailed information of the *PEBP* genes of *M. integrifolia.* Three different colors indicate three different subfamilies. The unit of molecular weight is kDa.

Gene Subfamily	Gene ID	Transcript ID	Sequence ID	Number of Amino Acid	Molecular Weight (MW)	Theoretical Isoelectric Point (PI)	Grand Average of Hydropathicity (GRAVY)
*MFT-like*	*MiMFT1*	*MiMFT1*	XP_042502857.1	172	18.98	9.16	−0.058
	*MiMFT2*	*MiMFT2*	XP_042520767.1	180	19.69	9.42	−0.236
	*MiMFT3*	*MiMFT3.1*	XP_042509881.1	203	22.66	5.87	−0.158
		*MiMFT3.2*	XP_042509888.1	172	19.03	7.02	−0.132
*FT-like*	*MiFT1*	*MiFT1.1*	XP_042482853.1	190	21.73	9.4	−0.464
		*MiFT1.2*	XP_042482854.1	183	20.83	9	−0.483
	*MiFT2*	*MiFT2*	XP_042494244.1	174	19.71	6.12	−0.402
	*MiFT3*	*MiFT3.1*	XP_042503696.1	174	19.74	7.95	−0.304
		*MiFT3.2*	XP_042503697.1	174	19.8	9.41	−0.347
	*MiFT4*	*MiFT4*	XP_042516980.1	185	20.96	7.68	−0.263
	*MiFT5*	*MiFT5*	XP_042517052.1	183	20.983	8.49	−0.425
*TFL1-like*	*MiBFT1*	*MiBFT1*	XP_042506411.1	173	19.58	9.47	−0.279
	*MiBFT2*	*MiBFT2*	XP_042510995.1	174	19.62	7.89	−0.352
	*MiBFT3*	*MiBFT3*	XP_042511022.1	207	23.12	9.41	−0.286
	*MiTFL1-like*	*MiTFL1-like*	XP_042489543.1	172	19.42	9.18	−0.273
	*MiTFL1*	*MiTFL1*	XP_042494365.1	172	19.44	8.66	−0.281

## Data Availability

Data are contained within the article or Appendix A.

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
