# Peer review of "Genome-Wide Characterization of PEBP Gene Family and Functional Analysis of TERMINAL FLOWER 1 Homologs in Macadamia integrifolia"

_plants, 2023, doi:10.3390/plants12142692_

Round 1
Reviewer 1 Report
The research and methods are presented clearly. A few minor edits are suggested below.
L118 search not searche
L146 delete first mention of phylogenetic
L183 replace is with are
L203 by confocal microscopy. Delete “the”
L213 specify from what cultivar of macadamia the samples were taken. Also, in the discussion please explain if samples from other cultivars could be expected to yield different results or not.
L213 specify what are “normal growth conditions”. Perhaps say under commercial field conditions.
L216 replace vegetable with vegetative
Fig 6 consider using the terms raceme and floret
L234 delete “even”
L238 Last sentence in this line is more appropriate in the discussion section.
L319 overexpressed not overexpressing
L329-332 Need to add more information in your discussion about how your research on PEBP genes will be used by breeders. Suggest what are the next steps for macadamia and give examples of these next steps being taken in other crops and the commercial outcomes.
L335 instead of “well managed” you should describe the management or provide a reference which does.
Reviewer 2 Report
The manuscript addresses an important issue regarding the characterization of the PEBP gene family in Macadamia integrifolia species and functional analysis of the Terminal Flowering 1 (TFL1) homologs. The manuscript is well structured, and the experimental plan and analysis are clearly presented, elaborating a genome wide characterization of PEBP gene family using current molecular methods to asses the function of the two TFL1 homologues found in the particular species. There are some minor issues that the authors should elaborate to improve the manuscript. These are as follows:
1. In the title the term “ homolog” should be in plural ‘homologs” as the study assessed two homologues the MiTF1 and MiTF1-like in M. integrifolia.
2. In the Introduction section it would be nice to indicate that the species Macadamia integrifolia is a dicot with genome (2n=28) size of 652-896Mb, (i.e., in ln 103-104) as in Figure 1 and in ln 136-138 the localization of important motifs is shown on chromosomes. You may indicate the exact genome size of the particular species.
3. In Figure 2 the initials of the plant’s species used in gene annotation on the dendrogram should be included in the figure legend. (i.e. Arabidopsis thaliana : At; Zea mays : Zm; etc.)
4. In figure 8 the parentheses with the icon initials (A,…S, T) would be better following the sentence explaining what they are, than preceding.
5. In the Discussion ln 278-279 should be elaborated regarding the status of the PEBP gene family in dicots as the species explored in the study, M. integrifolia, is a dicot.
6. Some typos and grammar errors i.e. ln 119 “searche”, should be “searches”. Ln 202 “were distributed overlap with both the nucleus”, should be “were distributed both in nucleus…”. Ln 273-274 “members in the M. integrifolia were identified” should be “members were identified in the M. integrifolia,..”. Ln 313 “All these studies indicate the comparatively conserved function” should be “All these comparative studies indicate the conserved function “. Ln 318 “may plays” should be “may play”. Ln 324 “Artificially cultivated” artificially should be deleted.
7. Ln 295-296 should be rephrased to clearly indicate what is meant.
Some typo and grammar errors should be elaborated.
